# Investigating Two Modes of Cancer-Associated Antigen Heterogeneity in an Agent-Based Model of Chimeric Antigen Receptor T-Cell Therapy

**DOI:** 10.3390/cells11193165

**Published:** 2022-10-09

**Authors:** Tina Giorgadze, Henning Fischel, Ansel Tessier, Kerri-Ann Norton

**Affiliations:** Bard College, 30 Campus Road, Annandale-on-Hudson, NY 12504, USA

**Keywords:** computational model, agent-based model, tumor heterogeneity, triple-negative breast cancer, cancer stem cells, CAR T-cells, immunotherapy

## Abstract

**Simple Summary:**

Chimeric antigen receptor (CAR) T-cell therapy has shown much promise in liquid tumors but often fails in solid tumors. This work uses a computational model to examine under what conditions this therapy might fail or be successful. The model includes interactions between cancer cells, CAR T-cells (treatment), and vascular cells (that feed and support tumor growth). From our results, we determined specific tumor conditions in which CAR T-cell therapy is predicted to fail and suggest a combination treatment that might improve the efficacy of the treatment.

**Abstract:**

Chimeric antigen receptor (CAR) T-cell therapy has been successful in treating liquid tumors but has had limited success in solid tumors. This work examines unanswered questions regarding CAR T-cell therapy using computational modeling, such as, what percentage of the tumor must express cancer-associated antigens for treatment to be successful? The model includes cancer cell and vascular and CAR T-cell modules that interact with each other. We compare two different models of antigen expression on tumor cells, binary (in which cancer cells are either susceptible or are immune to CAR T-cell therapy) and gradated (where each cancer cell has a probability of being killed by a CAR T-cell). We vary the antigen expression levels within the tumor and determine how effective each treatment is for the two models. The simulations show that the gradated antigen model eliminates the tumor under more parameter values than the binary model. Under both models, shielding, in which the low/non-antigen-expressing cells protect high antigen-expressing cells, reduced the efficacy of CAR T-cell therapy. One prediction is that a combination of CAR T-cell therapies that targets the general population of cells as well as one that specifically targets cancer stem cells should increase its efficacy.

## 1. Introduction

Cancer is characterized by the uncontrollable expansion of a subject’s own cells and several hallmarks have been identified that must occur for cancer progression and metastasis [1,2]. While advances have been made in the treatment of several cancer subtypes [3,4], many cancers still remain difficult to treat and recurrences remain frequent [5]. One such cancer is triple negative breast cancer (TNBC), which is highly aggressive and resistant to therapy [6]. This resistance is partially due to the fact that it lacks three common cellular receptors that are targeted in treatments, those being: estrogen, progesterone, and human epidermal growth factor receptor 2 (HER2) [7]. TNBC makes up about 10% of breast cancers in women, with approximately double this proportion among non-Hispanic Black women [8]. Recently, immunotherapy and combination immunotherapies have shown some success in treating the disease [9]. Specifically, chimeric antigen receptor (CAR) T-cell therapy has been shown to increase survival in patients with triple negative breast cancer [10,11].

Chimeric antigen receptor (CAR) T-cell therapy is a relatively new immunotherapy that allows for personalized treatment of cancer [12]. The T-cells are called chimeric since they have receptors incorporating an intracellular CD3ζ-activating domain with cytotoxic capabilities that trigger cellular apoptosis and an antigen-binding single-chain variable fragment with the ability to bind to cancer-associated antigens for the specific targeting of cancer cells [13]. CAR T-cell therapy has had varying levels of treatment success. It has been found that in some patients T-cells proliferate and expand substantially while in others this does not occur, limiting its effectiveness [13]. Other limitations are the difficulty in finding targetable cancer-associated antigens (CAA) and the heterogeneity of these cancer-associated antigens within the tumor [14]. For an antigen to be a successful target, it must not be expressed (or expressed very minimally) in normal tissue while also being expressed throughout the entire tumor [15]. This substantially limits the number of antigen targets that can be used. Furthermore, if the antigen is not sufficiently expressed in most cells of the tumor, it may lead to a large portion of the tumor being resistant to the therapy. Tumor heterogeneity leading to antigen expression heterogeneity is a major roadblock, especially, when only one antigen is targeted [14]. 

Recently, many computational models have focused on CAR T-cell therapy and immunotherapy (see [16] for reviews), and several recent agent-based models have focused on tumor immunotherapy [17,18]. Models looking at CAR T-cell proliferation found that high levels of CAR T-cell proliferation was able to completely eliminate the tumor [19], but another found it did not change the effectiveness of the therapy [20]. A Monte Carlo model of a CAR T-cell found that increases in surface receptor levels could actually decrease its ability to respond to stimulation [21]. Several other types of models have been used to study CAR T-cell therapy including a probability distribution model [22], a PK-PD model [23], and a control theory model [24]. Very few agent-based models (ABMs) have been used to study CAR T-cell therapy, but several have been used to examine immunotherapy. Agent based models of tumor immunotherapy have been used to examine a generic tumor model [25], glioblastoma [26], colorectal carcinoma [27], and lung cancer [28]. A recent paper used an agent-based model to study the effectiveness of CAR T-cell therapy in cell culture and tissue environments [29] but did not consider the tumor microenvironment.

This work builds an ABM to investigate CAR T-cell therapy in a breast cancer microenvironment and is expanded from our previous proceedings [30] to investigate a heterogeneous tumor cell antigen distribution in addition to a binary tumor cell distribution. The current ABM incorporates three modules based on previous works: the triple negative breast cancer (TNBC) module [31], an angiogenesis module [32], and a CAR T-cell module [33,34]. The TNBC module models the growth of a TNBC tumor, where the initial percentages of cancer stem cells and CCR5+/− cells and the migration rates of TNBC cells are based on in vitro experiments [31]. The effects of the percentages of CCR5+ and cancer stem cells on the growth of the tumor were investigated and it was found that while both treatments decreased tumor growth, without the complete elimination of cancer stem cells the tumor did not reach dormancy. The angiogenesis module investigated the individual and combined effects of endothelial cell migration and proliferation on tumor size and found that both over-proliferation and under-proliferation reduced the size of the tumor [32]. The angiogenesis module was also incorporated with the TNBC module and the inclusion of an immune cell module (macrophages) and fibroblasts allowed us to study tumor-stroma interactions [34]. A reduced version of this model was used to study cytotoxic T-cell immunotherapy [33]. 

An unknown of CAR T-cell therapy is what percentage of the tumor must express the antigen for the treatment to be successful [15]. Thus, one of the primary purposes of this paper is to look at different models of antigen expression and predict what percentage of antigens must be expressed for successful treatment, successful remission, or tumor stability compared to continual tumor growth. In this paper, we compare two different models of CAR T-cell therapy, a binary distribution of tumor cell antigens from previous work [30] to a new model with heterogeneous distribution of antigens. The previous model examined CAR T-cell therapy in which each TNBC cell was either susceptible or immune to CAR T-cell elimination (thus a binary distribution). In the current paper, we examine a more nuanced distribution of antigens, where each cell has a particular antigen level, which is the probability it would be eliminated by a nearby CAR T-cell. Here, we examine how the distribution of antigen levels in the tumor effect overall treatment and compare it to the binary distribution model. 

## 2. Materials and Methods

We have developed a new CAR T-cell therapy ABM with heterogenic (gradated) antigen presentation; this model is based on previous models of triple-negative breast cancer (TNBC), angiogenesis, and immunotherapy [31,32,33,34] and is an expanded version of CAR T-cell therapy with binary antigen presentation [30]. Agent-based models simulate elements in a model (such as cancer cells) as agents that have their own set of characteristics and individual behaviors. Agent-based cancer models try to simulate the localized behavior of every cell in the tumor microenvironment, such that the behaviors of the tumor as a whole will emerge. One of the limitations of agent-based modeling is that it is very difficult to model every single element that has an impact on the biological system being simulated; however, substantial results can be gleaned by simulating several of the most relevant actors.

The simulations are on lattice, which take place on two different grids, a cellular and a vascular grid. The cellular grid has the size of 50 × 50 × 50, where each voxel represents a space of 20 μm^3^, while the vascular grid has the size of 500 × 500 × 500, where each voxel represents a space of 2 μm^3^. An initial tumor of 100 cancer cells (80 progenitor and 20 stem cells) is placed in a corner of the grid. The cancer cells cannot leave the boundary of the grid, simulating a small tumor growing on the surface of normal tissue. The vasculature is initialized by putting 8 capillaries on the YZ plane, where they remain fixed. Each capillary consists of individual segments and has the ability to branch and become a sprout. We also assume that there is normal vasculature on the XY plane providing oxygen to the tumor. This initial setup is adapted from a previous model; for more information see [33].

### 2.1. Cancer Cell Module

The TNBC cell module (cancer cell module) is built off previous modeling work done by Norton et al. [34] and incorporates cancer in vitro/in vivo experimental data from triple-negative breast cancer cell lines. The module includes stem cells and progenitor cells which migrate, proliferate, senesce, die, and experience hypoxia. Each cancer cell is either CCR5+ or CCR5−, which governs its rate of migration and is based on experimental data [31]. Specifically, a CCR5+ cell can move 10X faster than a CCR5- cell (CCR5+: 8.3 μm/hr).

Each cancer cell in the simulation goes through several checks based on a decision tree to determine the actions they can take; see Figure 1. In short, each cell checks for quiescence, checks for hypoxia, checks for migration, checks for proliferation, checks for senescence, and then checks for apoptosis. To determine whether it becomes quiescent, it checks the 26 adjacent spaces (Moore neighborhood) to see whether there is any available space. If all adjacent spaces are occupied by other cells, the cancer cell becomes quiescent and does nothing for the rest of the iteration. To determine hypoxia, it checks whether the distance from the nearest mature vessel is greater than 200 μm, and if so, it is considered hypoxic (for more information, see the angiogenesis section). 

Assuming a free adjacent space is found, the cell will be able to migrate. The rate of migration is determined by several conditions. In vitro experiments showed that hypoxia increased the migration rate of MDA-MB-231 cells, a triple-negative breast cancer cell line, three-fold, whereas it decreased their proliferation rate by half [35]. Thus, if a cell is hypoxic, the rate of migration increases 3-fold. A CCR5+ cell’s migration rate increases 10-fold from the base CCR5− cell (0.83 μm/hr) as well. After migration, a cell can proliferate if there is available space. Stem cells have a base proliferation rate of 0.2/day and progenitor cells have a base proliferation rate of 0.5/day but hypoxic cells have half their base rate [31]. Progenitor cells always divide symmetrically, whereas stem cells have a 5% chance to divide symmetrically into another stem cell. Otherwise, the stem cell divides asymmetrically into a progenitor cell. Progenitor cells can only divide if they have not reached their division limit (12 divisions). Daughter cells are randomly placed into one of the available adjacent spaces and inherit the division limit and the antigen presentation of the parent. Each cell has a 5% chance of being CCR5+. Cells that have reached their division limit become senescent and have a 10% probability of dying each day. 

### 2.2. Vasculature Module

The angiogenesis module (vasculature module) is a slightly modified version of that in [34]. As in [30], the angiogenesis grid size has been reduced to (500 × 500 × 500) with a voxel size of 2 μm. The initial vasculature consists of 8 mature vessels on the two edges closest to the initial placement of the tumor; see Figure 2. Sprouting angiogenesis is caused by the release of vascular endothelial growth factor (VEGF) from hypoxic cells; therefore, the initial sprout does not form until after the first cell becomes hypoxic in the simulation. The sprout initially consists of a tip cell, which then proliferates to form a stalk cell behind it, attached to the initial mature vessel. After the first stalk cell is created the tip cell no longer proliferates and can only elongate and migrate. Tip cells migrate towards hypoxic cancer cells, which are a source of VEGF. The stalk cell proliferates, creating a quiescent phalanx cell where the stalk cell previously was, a stalk cell where the tip cell previously was, and a new tip cell at the head of the sprout extending in the same direction as the previous tip cell. A sprout can become mature when it either anastomoses with another tip cell or with a phalanx cell. If two sprouts are near one another they fuse to form a connect mature blood vessel. In addition, if a tip cell encounters another vessel, the two fuse together to form a mature vessel. Mature vessels are then assumed to release oxygen, causing nearby tumor cells to be normoxic. 

The angiogenesis module and the TNBC tumor module interact with each other through hypoxia, areas of low oxygen. TNBC cells that are greater than 200 μm from a mature vessel or the XY plane (as we consider that to be the normal tissue) are considered hypoxic [36]. Initially, the mature vessels are the 8 initial vessels, but as the simulation continues, sprouts are formed and once they anastamose are considered mature. Thus, every iteration of each cell determines whether it is hypoxic. Once a cell has been hypoxic for too long, it becomes anoxic and susceptible to apoptosis; thus, in the simulation we kill and remove any cell that has been hypoxic for more than 40 iterations.

### 2.3. CAR T—Cell Module

Before discussing the differences between the two different CAR T-cell models, we first discuss the components they have in common. In both CAR T-cell models, the initial setup and the migration module are the same, and every iteration, each individual CAR T-cell has the chance to migrate, kill nearby tumor cells, proliferate, and/or undergo apoptosis. Similarly to our previous work, ten initial CAR T-cells are released from the mature vasculature after a delay [30]. This is done by randomly choosing a mature node in the vasculature and placing the CAR T-cell in an adjacent voxel, if there is space. CAR T-cells that have exited the vasculature are programmed to migrate; for further information see [30]. Since CAR T-cells are some of the fastest immune cells, averaging speeds of 6.35 μm min−1 in brain tissue [37], CAR T-cell are able to migrate 16 times each iteration. First, each CAR T-cell determines whether it has moved more than 16 times (*cmigrate* > 0), and if not, it checks whether it is close to a cancer cell. A cancer cell is considered close if it is within 10 voxels of a tumor cell, which is within 200 μm. If a cancer cell is not nearby, it moves randomly into one of the 26 neighboring voxels that are open (Moore’s neighborhood). If a cancer cell is nearby, it checks whether it is in an adjacent voxel, and if it is, it stops moving. If it is not next to a cell, it determines whether it can move towards the cancer cell (i.e., another cell is not it its way), and if so, it moves towards it. Otherwise, it moves randomly into a free space next to it. Once it has moved towards a nearby cancer cell, it decreases its migration count (*cmigrate)* by one. Once again, it checks whether it can continue to migrate and reruns the whole cycle until it runs out of possible moves.

CAR T-cells are not introduced into the simulation until after 150 iterations (or 37.5 days). CAR T-cells proliferate at a rate of 0.8/day and their proliferation duration is 12 days [38]. Each iteration, a CAR T-cell checks whether it can proliferate, and if so, it places a new CAR T-cell in an adjacent voxel. If there are no available spaces to place a new CAR T-cell, it cannot proliferate. A CAR T-cell can die due to random cell death (apoptosis) or due to reaching the end of its lifespan (senescence). Each CAR T-cell has a random death rate of 0.204/day [25], in which we determine probabilistically each iteration whether the cell will die and eliminate it from the simulation. Since the chance of CAR T-cell death is increased with increasing numbers of CAR T-cells, after the number of CAR T-cells reaches 60,000 cells, the death rate of CAR T-cells is increased using Michaelis-Menton kinetics: *deathrate* = ((0.225 − *cartDeath*) × *Ncart*)/(37,500 + *Ncart*) + *cartDeath*, where *cartDeath* is the CAR T-cell death rate and *Ncart* are the number of CAR T-cells currently in the simulation. The lifespan of CAR T-cells is 14 days (or 56 iterations), and after a CAR T-cell has been alive for 14 days, it dies and is removed from the simulation. While CAR T-cell exhaustion is known to occur, the rate is essentially negligible for the timescale of this simulation [25]. 

Heterogeneity of cancer-associated antigens (CAA) can be due to differing levels of expression on cancer cells or due to cancer cells not expressing it at all. Therefore, in this paper, we consider two different models of CAR T-cell antigen expression: gradated and binary. Heterogeneous gradated antigen expression is modeled by each TNBC cell having different CAA levels and is novel to this paper. Whereas binary antigen expression is modeled by each TNBC cell either expressing the CAA or not and is the same as in [30]. The binary model is simulated such that a CAA level of 1 indicated the cell expressed the antigen and could be killed by a CAR T-cell and a CAA level of 0 indicated it did not express the antigen and could not be killed by a CAR T-cell. In the gradated antigen model, this is modeled by each cancer cell expressing a different level of antigen between 0 and 1, such that the expression level determines the probability it will be killed by a CAR T-cell. In order to examine the effects of antigen heterogeneity, a TNBC cell’s antigen level is pulled randomly from a normal distribution with a mean *anthet* and a standard deviation of 0.1. The parameter space consists of four values of antigen heterogeneity (*anthet*): 0%, 12.5%, 25%, and 50%. These numbers represent the mean percent chance of a tumor cell having the antigen present in a normal distribution. Mean values higher than 50% are not included in this paper because most of the tumors get eliminated after this point, results not shown. In both models of antigen expression, cancer cells pass their antigen expression level (CAA) to their progeny during cell division.

Each iteration, a CAR T-cell can kill up to five cancer cells with a rate of 20 cancer cells/day [39]. Each CAR T-cell checks its neighboring voxels for TNBC cells, and if there are more than five TNBC cells, it attacks five random cells. If there are five or fewer cells, the CAR T-cell attacks all of its neighboring TNBC cells. In the binary distribution, the CAR T-cell successfully kills a TNBC cell if its antigen cell level is 1 and does not kill it if its antigen level is 0. In the heterogeneous antigen model, the CAR T-cell kills its neighboring TNBC with a probability of *anthet,* the cell’s antigen level. Thus, if the TNBC cell’s antigen level is 0.20, the CAR T-cell has a 20% chance to kill it. One difference between the gradated and the binary model is that in the gradated model a CAR T-cell can attack the same cancer cell multiple times during an iteration. 

### 2.4. Statistics

In order to statistically compare metrics across the antigen presentation parameter space, we compared the resulting metrics at the last day of the simulation (day 75) of the 6 different runs for each antigen presentation value using one-way ANOVA to determine whether there was any statistical significance between the different antigen presentation values. If the ANOVA test was statistically significant, we determined which antigen presentation values were statistically significant from the others using a post-hoc Tukey test.

## 3. Results

### 3.1. Both Models of CAR T-Cell Therapy Can Successfully Reduce the Size of Tumor

One of the important questions governing this work is whether CAR T-cell therapy is effective at reducing or eliminating TNBC tumors. In both models, ten initial CAR T-cells are introduced through random points along the mature vasculature halfway through the simulation—at 37.5 days—and the therapy is monitored for another 37.5 days. Representative examples of tumor progressions over time with successful tumor reduction for each model can be seen in Figure 2. In both models, the number of cancer cells grows rapidly in the first half of the simulation in the parameter space without interference and then is reduced by CAR T-cell therapy for the next 37.5 days. In the gradated model (see Figure 2a), at day 50 the CAR T-cells can be seen expanding near the vessel they had infiltrated from. The CAR T cells have substantially killed the cancer cells by 62.5 days and only a small group of TNBC cells are left at day 75. In the binary model at 75% antigen expression (see Figure 2b), there is a similar trend of tumor elimination as in the gradated case at 12.5%. Note that in the binary model, Figure 2b at 75 days may look as though the tumor is eliminated but there are several living cancer cells scattered throughout the tumor. In both cases, once the CAR T-cells are introduced they substantially reduce the size of the tumor. 

### 3.2. The Distribution of Antigen Presenting Cells Effects the Rate of Tumor Reduction in Both Antigen Distribution Models

Since CAR T-cell therapy was shown to be able to successfully reduce tumors, we wanted to determine how the distribution of antigen presenting cells affected the rate of tumor reduction. Figure 3 (NumberofCells) shows the number of cells over time across the parameter space in both binary and gradated antigen models. In the next section, we discuss the gradated heterogeneity model, and in the section after that, we briefly summarize our results from [30] and discuss how the two models differ.

#### 3.2.1. Gradated Heterogeneity Antigen Model

In the gradated heterogeneity antigen model, each cancer cell is assigned a different expression level of cancer-associated antigen (CAA), such that the level is the probability of being killed by a CAR T-cell. In the heterogeneity model, we sample these probabilities from a normal distribution with a given mean (0%, 12.5%, 25%, 50%) and a standard deviation of 0.1 to assign each cell a level of cancer-associated antigen (CAA). At 0% mean antigen expression (yellow), the number of cells grows rapidly in the first half of the simulation for all values. Once CAR T-cells are introduced at day 37.5, the number of cells keeps increasing but the rate of growth is reduced as a result of the treatment; see Figure 3a (NumberofCells). The slope of the yellow plot starts to plateau around day 62.5, indicating that the CAR T-cells were able to slow down the rate of growth of the tumor, even if the therapy failed at reducing the overall size. Most tumor cells have 0% chance of having the antigen, and thus, most cells are immune to the treatment, therefore, the cells that are killed by CAR T-cells reduce the overall growth of the tumor without actually shrinking it. The 12.5% mean antigen expression tumor starts to decrease more abruptly than the 0% tumor. Unlike the latter, the size of the tumor is actually reduced as opposed to simply slowing down its progression. At 12.5% mean antigen expression, the number of cells goes down from 3000 to 1500 cells between days 37.5 and 75. In the 25% and 50% cases, the tumor actually gets eliminated and the simulation ends at around day 62.5 with no cancer cells left. This is an important result, as this demonstrates the region in the parameter space where tumor reduction goes to tumor elimination. We conducted a one-way ANOVA and post-hoc Tukey test on the number of cancer cells over time for the gradated antigen model and found that the number of cancer cells is statistically different across the entire parameter space (*p* = 0.0011), with the 0% group being significantly different from the other three. Therefore, as long as the tumor is somewhat susceptible to the treatment, the therapy will be effective at reducing the size of the tumor.

#### 3.2.2. Binary Heterogeneity Antigen Model

Figure 3b (NumberofCells) shows the number of cancer cells over time averaged over 6 runs. Similar to the gradated model, the size of the 25% antigen expression tumor (yellow) in the binary model does not get reduced as a result of the treatment. The number of cells grows rapidly in the first half of the simulation, and after that point, the number of cells starts to decrease in all of the parameter conditions except at 25% antigen presentation. The number of cancer cells continues to increase slowly. Even though most cells are immune to treatment, the ones that have the antigen do get killed, reducing the number of cancer cells up to 25%. At 50% antigen presentation, there is a clearer reduction of the size of the tumor, followed by a faster reduction at 75% antigen presentation. At 25% and 50% mean antigen expression, the differences in the tumor reduction rates are much more noticeable compared to the gradated model. Although the two plots have similar shapes, the 50% (light blue) plot reduces slower than the 75% (red) plot. At 98% antigen presentation, we see that most cells have been eliminated. Very few cells survive at 98% antigen presentation, all of which do not present the antigen. This is visualized in Figure 4b (column 4), where the remaining tumor at 98% antigen expression is all red (antigen non presenting). A one-way ANOVA test on the number of cancer cells at 300 iterations (or 75 days) and a further post-hoc Tukey test showed that 98% was significantly different from the other three groups [30]. After day 62.5, both 98% and 75% antigen expression tumors stop decreasing in size. This occurs because at this point most of the remaining cancer cells are immune to the treatment, so the CAR T-cells have fewer targets to eliminate and the tumor reduction rate plateaus. 

In comparison of the two models, there is a positive trend between the number of antigen-presenting cells in tumors and their reduction rates. The gradated antigen heterogeneity model is similar to the binary one but the differences across the parameter space are less pronounced. These findings suggest that in the gradated model, there is less difference between the reduction rates of tumors above 25% mean antigen expression. In both cases, CAR T-cells were able to reduce the size of cancer at almost the same rate and eliminate the entire tumor at about the same time. Thus, in the heterogeneity model, there is still a positive trend between the number of antigen-presenting cells in the tumor and the rate of tumor reduction; however, this trend is less pronounced than in the case of the binary model. Therefore, while the size of the tumor gets reduced in all cases except the first in both models, we see that the binary therapy fails to fully eliminate the tumor as cells survive until the end of the simulation in all cases. This is an important result, as the CAR T-cell therapy was able to fully eliminate tumors in some cases in the gradated model. Thus, the therapy is more effective in the gradated heterogeneity model. The survival of antigen non-presenting cells is likely to cause a relapse of the tumor over time. 

### 3.3. There Is a Positive Trend between the Number of Antigen Presenting Cells and the Growth of CAR T-Cells in Both Models

After showing that the treatment affects the size of tumor and tumor reduction rates, we wanted to determine whether the number of antigen-presenting cancer cells affects the growth of CAR T-cells in the simulation. At the end of each run, we collected the number of CAR T-cells over time (every 1 iterations = 0.25 days); see Figure 3b (NumberofCART). Here, the effect of the percentage of antigen-presenting cells on the number of CAR T-cells in both models are similar and so we will discuss them together. The ordering of the plots is the same; the number of CAR T-cells increases throughout the simulation and across parameter spaces in both models. The yellow plots (0% mean antigen expression tumor in gradated and 25% antigen expression in binary) increase until they reach about 15,000–18,000 CAR T-cells in both models. This is followed by light blue (12.5% mean antigen expression in gradated and 50% antigen expression in binary), which reaches 20,000 CAR T-cells in the binary model and 30,000 CAR T-cells in the gradated model. Comparing Figure 3 (NumberofCells), the size of the tumor at 50% antigen expression in the binary model is on average higher than the 12.5% mean antigen expression (light blue) plot in the gradated model. This suggests that there is less space for CAR T-cells to grow and spread in the binary model than the gradated model, and thus, the number of CAR T-cells only reaches 20,000 compared to 30,000 in the gradated model. This is also shown in Figure 4b at 50% binary antigen presentation; it is clear that a substantial number of tumor cells survive (both antigen and non-antigen presenting). The 25% mean antigen expressions in the gradated and 75% antigen expression in the binary model (red) plots have similar growth patterns, but the gradated plot outgrows the binary one as well. Similarly, examining the sizes of tumor in each model, having more cancer cells present in the binary model leaves less room for CAR T-cells, and thus, the number of CAR T-cells is higher for the gradated model. Similar trends happen with 50% mean antigen expression in gradated and 98% antigen expression in binary model (dark blue) plots, but in higher presenting tumors (red and dark blue plots), the growth rates of CAR T-cells are higher for the gradated model than the binary model. In both models, the number of CAR T-cells reaches around the same maxima, but the tumors get eliminated before the end of the simulation in the gradated plots. Thus, the growth rates for the red and dark blue plots are higher for the gradated model. Overall, there is a positive trend between the percentage of antigen presenting cells and the number of CAR T-cells in both models. CAR T-cells can only proliferate and spread if there is space, so if most cancer cells are immune to treatment, then there is less room in the grid for the CAR T-cells to spread. Hence, it is expected to see CAR T-cells grow at a slower rate with more therapy-resistant cancer cells. Furthermore, one-way ANOVA tests on the number of CAR T-cells over time showed statistical significance across parameter space for both models (*p* = 0.03 for gradated and *p* = 0.0077 for binary). Post-hoc Tukey tests showed that in the binary model, 25% and 98% antigen expression tumors were statistically different, but there was no significant difference among the other tumors. For the gradated model, the 0% mean antigen expression tumor was different from the other three groups, but there was no difference among the 12.5%, 25%, and 50% mean antigen expression of the tumors. This shows that in the gradated model, as long as the tumor is somewhat susceptible to the treatment and has more than 0% mean probability of presenting the antigen, CAR T-cells will increase sufficiently.

### 3.4. There Is a Positive Trend between the Percentage of Antigen Expression and the Number of Cancer Cells Killed by CAR T-Cells throughout the Simulation in Both Models of Antigen Distribution

CAR T-cell treatment reduced the size of the tumor in both models (except the 25% presenting tumor in the binary model and the 0% mean antigen expression tumor in the gradated heterogeneity model). Next, we wanted to verify that the increase in antigen expression levels increased the number of cancer cell deaths. We collected the number of cancer cell deaths over time for the different treatments, which are shown in Figure 3 (NumberofDeath).

#### 3.4.1. Gradated Heterogeneity Antigen Model

As expected, fewer cells died in tumors with lower mean antigen presentation compared to tumors with higher antigen presentation. Figure 3a (NumberofDeath) shows that the 0% antigen expression (yellow) plot stays mostly flat throughout the simulation, with a slight increase in slope at around day 40. Therefore, the cancer reduction rate is fairly slow in this case, which can also be seen in the yellow plot in Figure 3a (NumberofCells). Figure 4a column 4 shows that the surviving tumor is rather big and has not been decreased much as a result of the therapy. The 12.5% antigen expression tumor (light blue) plot in Figure 3a (NumberofDeath) increases up to about day 57, after which it starts to gradually decrease again. This is due to the fact that at this time the tumor has been reduced to the point that there are fewer antigen-presenting cells left for CAR T-cells to target, and thus, the number of cell deaths decreases. Similar trends are observed with the 25% and 50% mean antigen expression tumors; there is a much more noticeable jump in the number of cancer cells killed at around day 55. Thus, the therapy is immediately effective, and the number of cancer cells killed keeps going up until the tumor has been reduced to a point where there are very few cancer cells left to be killed. This demonstrates a positive trend between the rate of the tumor reduction and the mean antigen presentation value of the cancer cells.

#### 3.4.2. Binary Heterogeneity Antigen Model

The binary antigen model differs slightly from the gradated antigen heterogeneity model, Figure 3 (NumberofDeath). The yellow plots corresponding with the lowest percentage of tumors (25% antigen expression for binary and 0% mean antigen expression for gradated) both have very small slopes and are constantly low throughout the simulation. The light blue plots corresponding with 50% antigen expression in binary and 12.5% mean antigen expression in heterogeneity models also show similar trends. The differences between the two models occur in the remaining two antigen expression levels (dark blue and red plots). The red plot of the gradated model (25% mean antigen expression) has a much bigger jump than the red plot of the binary model (75% antigen expression). The average peak in the gradated model is around 800 and around 200 in the binary model. The dark blue plots (98% antigen expression in binary and 50% mean antigen expression in gradated) have a similar shape with similar maximums (at about 800 cells). It is also noticeable that the error bars are smaller in the binary model. In the gradated model, there is more overlap across parameter space, especially with 25% and 50% mean antigen expression tumors, both of which have very high cancer kill counts. Similarly, there is a positive trend between the number of antigen-presenting cells and the number of tumor cell deaths

### 3.5. In Both Models, There Are Parameter Values in which Cancer Stem Cells Get Eliminated

Previous agent-based models developed by our lab showed the importance of eliminating stem cells [33,40] in order to reduce the size of tumor as well as to reduce the risk of relapse. Therefore, we wanted to look at how the number of stem cells in the simulation were affected by CAR T-cell therapy.

#### 3.5.1. Gradated Heterogeneity Antigen Model

In the gradated model, the growth of stem cells for all four cancers is very similar until the CAR T-cells get introduced and the number of stem cells starts to drop at different rates depending on the tumor, Figure 3a (NumberofStems). The 0% mean antigen expression plot (yellow) drops at an extremely slow rate, slower than the yellow plot of the binary version. Eventually, it reaches 20 stem cells on average, which is how many the simulation started with, so the overall number of cancer stem cells was not reduced in the 0% mean antigen expression tumor but did not increase either. For the 12.5% mean antigen expression tumor, there is a more consistent yet relatively slow reduction until around 16 stem cells at the end of the simulation. The higher presenting tumors (25% and 50% mean antigen expressions) had stem cells being killed at higher rates. They both show a large drop that was not observed in the other tumors. In addition, 25% and 50% mean antigen expression tumors are the only ones in the gradated model where stem cells were completely eliminated. This is also confirmed by Figure 3a (NumberofCells), which shows that the tumor was completely eliminated with 25% and 50% mean antigen expression. In other words, there is a relationship between stem cell elimination and tumor elimination. To check for significance across the parameter space, a one-way ANOVA test was conducted on the number of stem cells over time, which showed a significant difference between antigen expression and stem cell numbers with a *p*-value < 0.01 (0.000001). The post-hoc Tukey test showed that there was no significant difference between 0% and 12.5% or 25% and 50%, but the two groups (0%, 12.5% and 25%, 50%) were significantly different from each other. As the tumors approach 25% or higher antigen presentation, stem cells have a chance of being fully eliminated, which is crucial for avoiding relapse.

#### 3.5.2. Binary Heterogeneity Antigen Model

Figure 3b (NumberofStems) shows the overall trends for stem cells over time for the binary antigen model. Once the CAR T-cells get introduced on day 37.5, numbers of cancer stem cells start to drop. Cancer stem cells in the 25% antigen expression plot barely change, only dropping to around 18 on average across the runs. The 50% and 75% antigen expression tumors are similar to each other, dropping to around 16 cancer stem cells at the end of the simulation. At 98% antigen expression, there is a considerable jump at day 50 followed by a drop in cancer cell deaths, with a few simulations resulting in stem cell elimination. Such a drop is similar to the drop with the 25% or 50% mean antigen expression tumor in the gradated version. In other words, while the number of cancer stem cells gets reduced across the parameter space, they only get eliminated in 98% antigen expression tumors. This is due to the fact that cancer stem cells are a small population in the tumor. Therefore, statistically speaking, eliminating all the stem cells would require a very high antigen-presenting tumor. A one-way ANOVA and post-hoc Tukey test showed that the 98% antigen expression tumor was significantly different from other groups, but there was no significance between the 25%, 50%, and 75% antigen expression tumors [30]. Thus, in order to prevent tumors from relapsing after CAR T-cell therapy, they require a rather high percentage of antigen-presenting cancer cells. Otherwise, the unlimited replicative ability of cancer stem cells would facilitate a relapse of the tumor over time. Thus, the gradated CAR T-cell therapy proved more effective since the 25% mean antigen expression tumor had high enough antigen presentation to fully eliminate stem cells, whereas in the binary version a presentation of 98% was required. 

### 3.6. In Both Models, Antigen Presenting Cells Can Form a Shield over Antigen Non-Presenting Cells That Blocks CAR T-Cells from Killing Vulnerable Cancer Cells

In both the gradated and binary heterogeneity models, some antigen-presenting tumors exhibited a phenomenon we call “shielding”. Shielding happens when a layer of antigen non-presenting/low antigen-presenting cells forms a protective wall over antigen-presenting cells that prevents CAR T-cells from accessing and subsequently killing those vulnerable cells. This results in the survival of antigen-presenting cells that would normally be eliminated by the treatment. This shield blocking of the CAR T-cells’ access to antigen-presenting cells is a challenge, as portions of tumor that could have been tackled by the therapy never get accessed, which renders the therapy less effective. This may partially explain why CAR T-cell therapy has been effective in liquid tumors but less so in solid tumors. With the binary model, we saw examples of shielding in 50% and 75% antigen expression tumors; Figure 5 shows an example of shielding in both models. Shielding was not observed with either the lowest or the highest presenting tumors (lowest being 0% mean antigen expression for gradated and 25% antigen expression for binary and highest being 50% mean antigen expression for gradated and 98% antigen expression for binary). This occurred because lower presenting tumors mostly consist of antigen non-presenting cells and have too few antigen-presenting cells to form a shield over them. Conversely, higher presenting tumors mostly get eliminated, so there are too few remaining treatment-resistant cells to form any kind of shield. The reason why shielding happens is likely due to the fact that CAR T-cells kill most antigen-presenting cells while there are no more antigen-presenting cells nearby, thus creating a layer of resistant tumor cells. By preventing CAR T-cell therapy from reaching its full potential in reducing the size of cancer, shielding could facilitate the relapse of the tumor later in time.

## 4. Discussion

Previous in vitro works have shown that CAR T-cells encoding an MSLN-targeted and Myc-tagged scFv were significantly effective at killing a MBA-MD-231 cancer cell line (a type of TNBC) [41]. Thus, we have developed an agent-based model of two distinct models of CAR T-cell therapy and triple-negative breast cancer antigen expression. There are two difficulties to overcome in selecting a cancer-associated antigen clinically or experimentally. The first is whether one can find an antigen that is expressed enough on the cancer cells of a particular patient’s tumor to be effectively treated with CAR T-cell therapy. Therefore, we examined what percentage of the tumor needs to express an antigen in the binary model and what expression level each tumor cell must have in the gradated model for effective CAR T-cell treatment. We found that tumor elimination occurred in the heterogeneous model when antigen expression levels were 25% or greater but did not occur in the binary distribution unless the levels were close to 100%. These results can help predict how successful a specific CAR T-cell therapy might be clinically. By measuring the percentage of a patient’s cancer cells that express the specific antigen (CAA) and/or measuring the percentage of a patient’s cancer cells that respond to a specific CAR T-cell therapy, we can use this model to help predict the efficacy of the treatment. The percentage of antigen-expressing cancer cells would roughly correspond to the percentage of antigen-presenting cancer cells in the binary model, whereas the cancer cell response to CAR T-cell therapy would correspond to the mean antigen presentation in the gradated model. Thus, you could use these cutoffs in the parameter spaces to predict how successful the CAR T-cell treatment might be for a specific patient. 

From our results, it is clear that having even a small percentage of cells that are completely resistant to the CAR T-cell therapy can substantially reduce its efficacy. Therefore, finding an antigen that is present in all cancer cells is critical for CAR T-cell therapy. This seems to be a difficult task as a good antigen (mesothelin) was only present in 50% of cells in patients with TNBC [42]. In a different study, IAM-1-expressing CAR T-cells killed over 50% of TNBC cells in vitro, which was considered a success [43]. In addition, even in tumors with fairly uniformly expressed antigens, the tumors can escape immunotherapy and lose the antigen that was selected for [44]. The survival of cells that are antigen non-presenting has been shown to be a major cause of tumor recurrence [45].

A second difficulty is finding an antigen that is expressed significantly across breast cancer or TNBC tumors. In a screening of primary breast tumors, one study found that MSN was expressed at a low level in 65% of patients and at a high level in 19% of patients [42]. In another study, CSPG4 was expressed in about 73% of primary TNBC lesions [46]. We do not address finding an antigen expressed across different types of TNBC here specifically, except to show that if a TNBC tumor does not express the antigen, CAR T-cell therapy is ineffective, although one benefit of CAR T-cell therapy is that it can be tailored for each individual patient and so an antigen could be identified for the specific tumor in question.

These simulations showed that antigen non-presenting/low antigen-presenting cells can form a shield and protect antigen-presenting cells from elimination. It has been discussed in several reviews that CAR T-cell infiltration into the tumor is a major roadblock for its success in solid tumors [14,47]. Various cell types have been found to prevent CAR T-cell infiltration into the tumor as well as a dense extra cellular matrix [14]. Two specific obstacles have been the limited ability of CAR T-cells to infiltrate into the tumor because of “cold” tumors that either do not effectively express the chemokines necessary to recruit CAR T-cells or due to the inability of the CAR T-cells to infiltrate through the tortuous tumor vasculature [48]. The architecture and outgrowth of the tumor itself has also been associated with the in/ability for TILs to infiltrate into the solid tumor [48]. Thus, combination therapies have been proposed to improve the effectiveness of CAR T-cell therapy in solid tumors [49]. For instance, another in silico model has found that CAR T-cell therapy with a combination of two antigens allowed for prolonged tumor control [50]. An important implication of these results is not only can the tumor structure and microenvironment have the ability to prevent CAR T-cell access to susceptible cancer cells, but low antigen-expressing cancer cells themselves can also prevent accessibility. The microenvironmental limitations of CAR T-cell therapy have not been explored specifically in this model but will be explored in future studies. 

Another outcome of this work is the finding that reduction of the stem cells is needed for effective CAR T-cell treatment. The correlation between the number of stem cells killed by immune cells and the reduction in the tumor size was shown in previous work [33]. Several recent reviews have proposed using CAR T-cells to specifically target cancer stem cells in solid tumors [51,52]. Several cancer stem cell antigens have been found to be successful targets for CAR T-cell therapy, such as CD133, EGFRvIII, and high molecular weight melanoma-associated antigen [53]. One challenge of CAR T-cell therapy targeting cancer stem cells is finding a tumor-associated antigen (TAA) that is found on the surface of cancer stem cells and not normal stem cells [54]. Another issue with targeting cancer stem cells is that they are a small population of cells, and thus, as discussed above, if non cancer stem cells are not targeted, they will form a “shield” and protect the cancer stem cells from being found and eliminated by the CAR T-cells. Therefore, we propose combination therapies that specifically target the bulk of the tumor to reduce the shielding effect as well as therapies that specifically target the cancer stem cells that are hidden within the bulk of the tumor. A possibility would be using SynNotch circuits that could first target the general tumor population and then target cancer stem cells. This is another area that we have not specifically explored in this paper but will be explored in future research. 

## 5. Conclusions

In this paper, we develop a novel agent-based in silico trial of CAR T-cell therapy using a heterogeneous distribution of antigens and compare it to a previous model with a binary distribution of antigens [30]. We found that the heterogeneous model reduced the size of the tumor based on the antigen distribution of the cancer cells, and in a small number of cases, the treatment was able to eliminate the tumor. The percentage of antigen expression was positively related to the number of CAR T-cells and the number of cancer cells killed by CAR T-cells. Interestingly, we found that cells without or with little antigen expression could shield cells with high antigen expression from being killed by CAR T-cells. These results give us insights into CAR T-cell therapy and its efficacy. The most successful treatments in both models were the ones that killed more stem cells. These results suggest that having a combination of CAR T-cell therapy that targets the bulk of the tumor with a CAR T-cell therapy (or other therapy) that specifically targets cancer stem cells might increase the efficacy of the treatment. 

## Figures and Tables

**Figure 1 cells-11-03165-f001:**
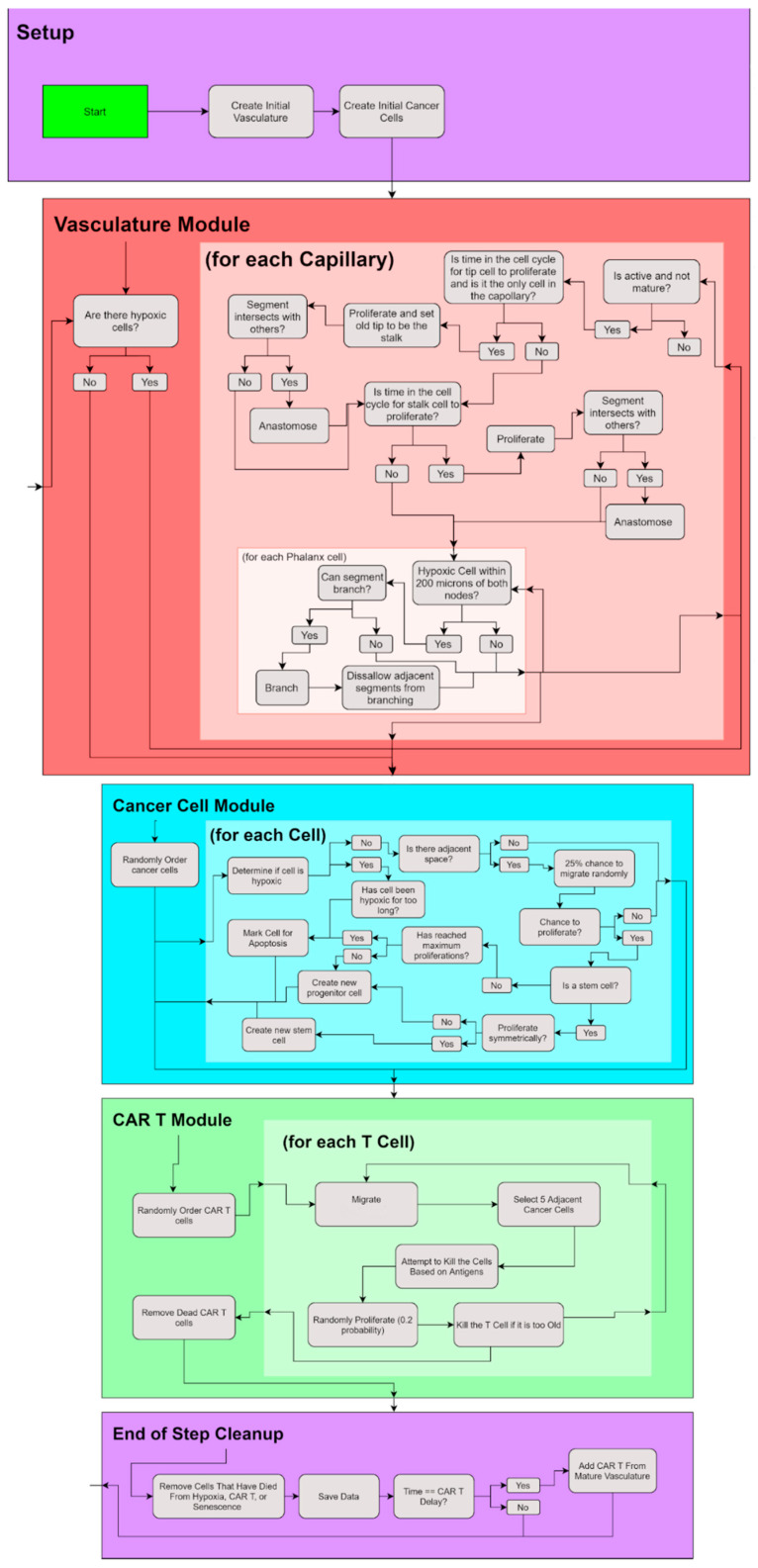
Flowchart of the CAR T-cell TNBC model.

**Figure 2 cells-11-03165-f002:**
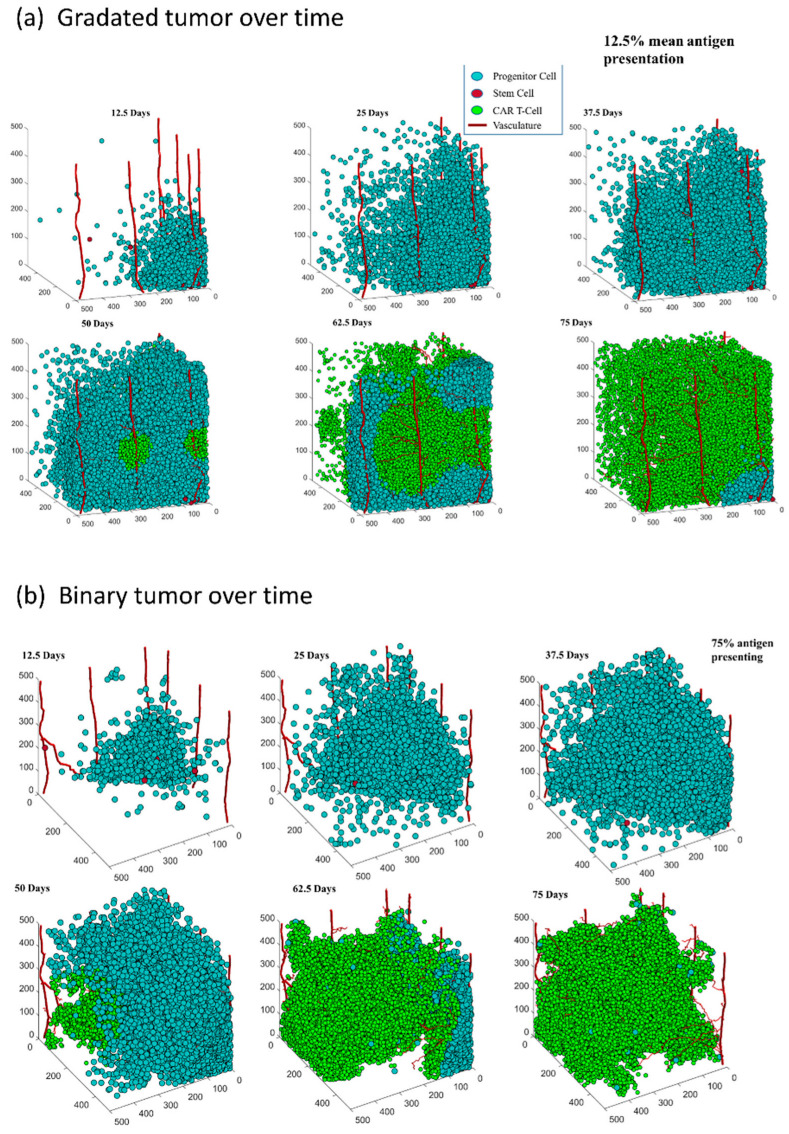
3D plots of tumors with gradated (**a**) and binary (**b**) heterogeneity plotted over time with 12.5-day intervals starting from day 12.5 and ending on day 75, the last day of our simulations. Tumor cells are plotted in blue, CAR T-cells are plotted in green, and the vasculature is plotted as red capillaries growing across the grid. The data in the gradated heterogeneity tumor progression (**a**) is from a run with 12.5% mean antigen expression tumor cells. The data in the binary heterogeneity tumor progression (**b**) is from a run with 75% antigen presenting tumor cells. CAR T-cells are introduced halfway through the simulation, on day 37.5.

**Figure 3 cells-11-03165-f003:**
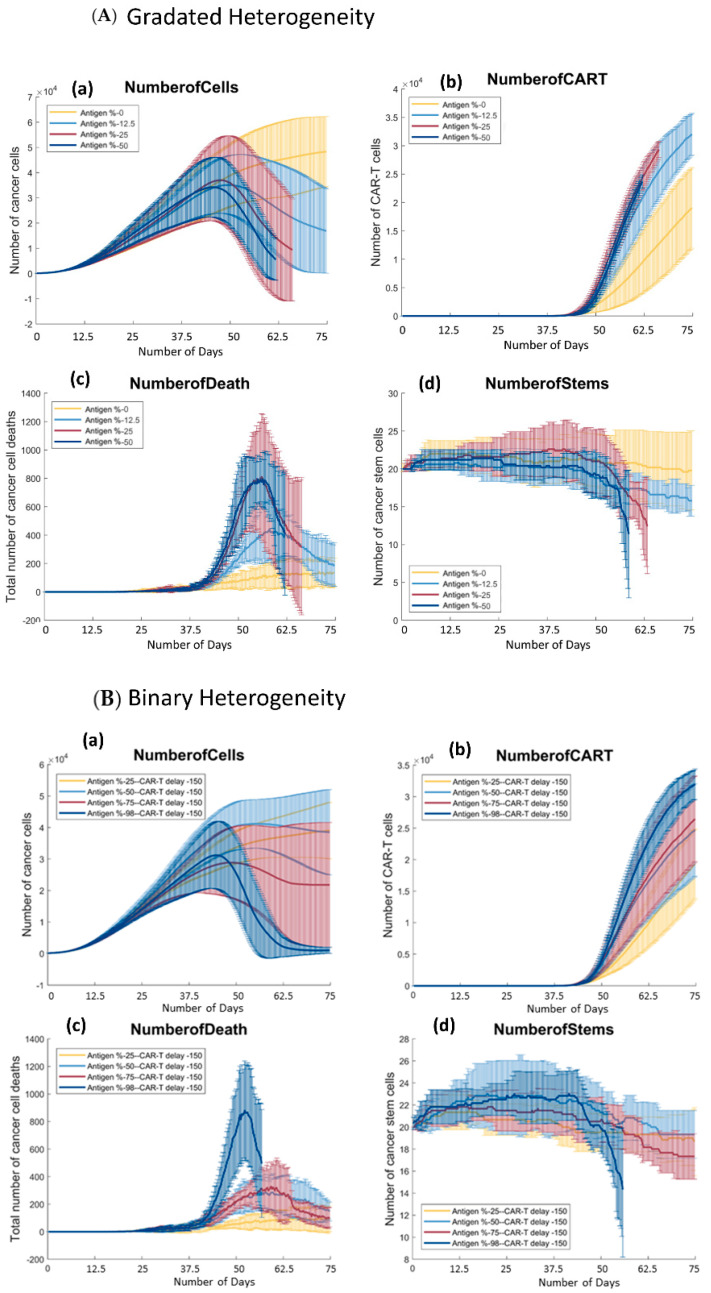
Tumor metrics over time for gradated heterogeneity (**A**) and binary heterogeneity (**B**). In both cases, the metrics include total number of tumor cells over time (**a**), total number of CAR T-cells (**b**), total number of cancer cell deaths (**c**), and total number of stem cells over time (**d**). Each iteration represents 6 h of real time. For the gradated heterogeneity metrics, 0% mean antigen expression is plotted in yellow, 12.5% mean antigen expression is plotted in light blue, 25% mean antigen expression is plotted in red, and 50% mean antigen expression is plotted in dark blue. For the binary heterogeneity metrics, 25% antigen presentation is plotted in yellow, 50% antigen presentation is plotted in light blue, 75% antigen expression is plotted in red, and 98% antigen expression is plotted in dark blue. Note: the parameters in which the line does not reach day 75 indicate that one of the tumors died out at this point and no further data was collected.

**Figure 4 cells-11-03165-f004:**
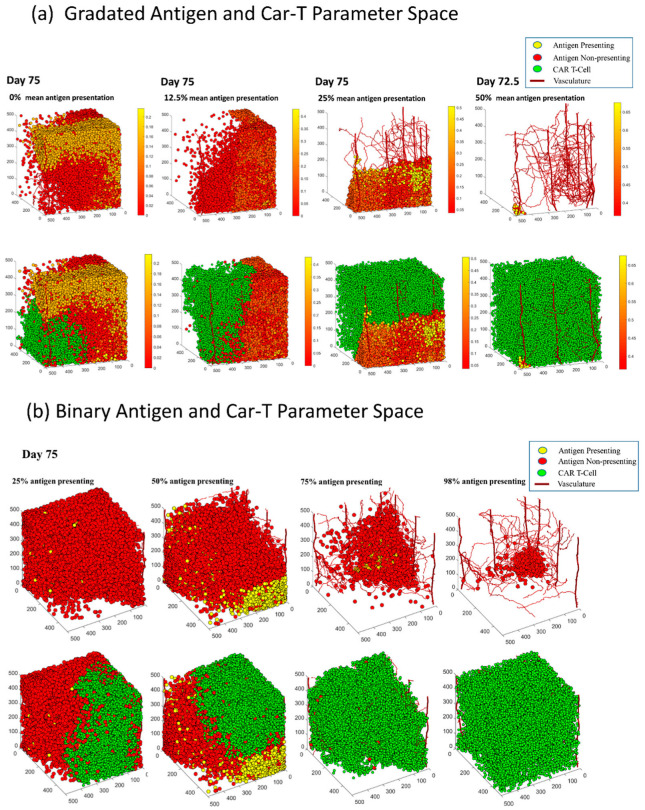
3D plots of tumors on the last day of our simulation, day 75, with different antigen presentations. For tumor with gradated heterogeneity (**a**), the data comes from 0%, 12.5%, 25%, and 50% mean antigen presentation (left to right). Vasculature is plotted as red capillaries growing across the plane. Tumor cells with 100% chance of presenting the antigen are plotted in yellow. Tumor cells with 0% chance of presenting the antigen are plotted in red. Every cell with a chance of antigen presentation between 0% and 100% is plotted on a gradient from red to yellow. CAR T-cells are plotted in green and are introduced on day 37.5 of the simulation. For the tumors with binary heterogeneity (**b**), the data comes from runs with 25%, 50%, 75%, and 98% antigen presentation. Tumor cells with 100% antigen presentation are plotted in yellow; cells with 0% antigen presentation are plotted in red.

**Figure 5 cells-11-03165-f005:**
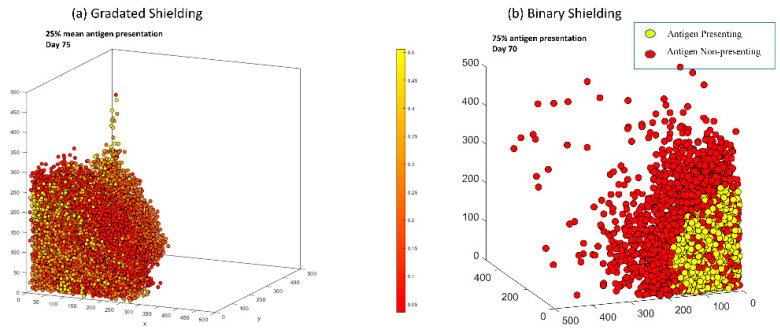
Examples of antigen shielding occurring in tumors with gradated heterogeneity (**a**) and binary heterogeneity (**b**). For gradated heterogeneity, the 3D plot shows tumor cells with 100% of antigen presentation in yellow, tumor cells with 0% antigen presentation in red, and tumor cells with intermediate probabilities of antigen presentation on a gradient. For binary heterogeneity, antigen-presenting tumor cells are plotted in yellow and antigen non-presenting tumor cells in red. In both cases, the figure shows a layer (shield) of antigen non/low-presenting cells forming a shield with high antigen-presenting cells. In both cases, such a shield makes it difficult for CAR T-cells to break through that layer, as the cells making up that layer are either fully or highly immune to CAR T-cell therapy.

## Data Availability

The dataset and code is provided on nortoncompbiolab.xyz/research.

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
