# Peer review of "Investigating Two Modes of Cancer-Associated Antigen Heterogeneity in an Agent-Based Model of Chimeric Antigen Receptor T-Cell Therapy"

_cells, 2022, doi:10.3390/cells11193165_

Round 1
Reviewer 1 Report
The manuscript by Giorgadze et al., entitled “Investigating Two Modes of Cancer Associated Antigen Heterogeneity in an Agent-Based Model of Chimeric Antigen Receptor T-Cell Therapy” presents an interesting work where the authors have developed a computational model to predict the efficacy of CAR-T cells in eliminating tumors by analyzing the distribution and expression of tumor associated antigens. The idea is very clearly conceptualized and the research is very interesting. The work has been well executed and the manuscript has been well written. However, the authors must address how this model can be a useful tool for decision making in choosing CAR-T cell therapy for patients with different cancers. Moreover, a clear description of the methodology with subheadings would be better for easy comprehension.
Author Response
Reviewer 1:
The manuscript by Giorgadze et al., entitled “Investigating Two Modes of Cancer Associated Antigen Heterogeneity in an Agent-Based Model of Chimeric Antigen Receptor T-Cell Therapy” presents an interesting work where the authors have developed a computational model to predict the efficacy of CAR-T cells in eliminating tumors by analyzing the distribution and expression of tumor associated antigens. The idea is very clearly conceptualized and the research is very interesting. The work has been well executed and the manuscript has been well written.
We thank the reviewer for their kind summary.
However, the authors must address how this model can be a useful tool for decision making in choosing CAR-T cell therapy for patients with different cancers.
We agree with the reviewer and have added a few sentences in the discussion section as to how the model might be useful for therapy response prediction. We have added the following paragraph:
“We found that tumor elimination occurred in the heterogeneous model when antigen expression levels were 25% or greater but did not occur in the binary distribution unless the levels were close to 100%. These results can help predict how successful a specific CAR T-cell therapy might be clinically. By measuring the percentage of a patient’s cancer cells that express the specific antigen (CAA) and/or measuring the percentage of a patient’s cancer cells that respond to a specific CAR T–cell therapy, we can use this model to help predict the efficacy of the treatment. The percentage of antigen expressing cancer cells would roughly correspond to the percentage of antigen presenting cancer cells in the binary model. Whereas, the cancer cell response to CAR T-cell therapy would correspond to the mean antigen presentation in the gradated model. Thus, you could use these cutoffs in the parameter spaces to predict how successful the CAR T-cell treatment might be for a specific patient.”
Moreover, a clear description of the methodology with subheadings would be better for easy comprehension.
We thank the reviewer for their comments and have added headings to the methods section for clarity.
Reviewer 2 Report
In this work, the authors present two improved multimodular in silico models to assess basic pathophysiological questions in CAR T-cell therapies for solid tumors, in this case triple negative breast cancer. One model uses a binary, the other a gradated model for antigen expression on tumor cells. Vascular, CAR T-cell as well as tumor modules are combined to gain insight into these questions. In the gradated model (which represents the reality of antigen expression more closely than the binary model) a cut-off of 25% is found to be critical for efficient tumor cell killing. More important, findings that antigen negative tumor cells can from a barrier to shield of CAR-T cells is, in my opinion, one the most important statements and should be discussed in detail.
Here a my comments:
Abtract: Well written, please check minor grammar mistake line 14 "to be fail".
Introduction:
The introduction is well written. However, depending on the intended readership, I would suggest paraphrasing or shortening the introduction in some aspects or try be more concise. For the readers of Cells your findings are of interest but the main narrative should focus on its translational importance in future preclinical or clinical studies for CAR T-cell therapies. Try to reduce presenting more and more computational models, concentrate on the most import ones (such as previous work of yours) and introduce (and later discuss) opportunities of your findings for future studies (Should biologists/clinician scientists focus on better CAR T-cell infiltration, how could this be achieved? Are your findings applicable in in vitro or even in vivo models?)
Lines 49-51: The explanation for chimeric is somewhat confusing. In my opinion the term chimeric in CARs rather describes the receptor incorporating both 1) antigen-recognition moieties and 2) intracellular T-cell activating domain that includes a CD3 zeta chain, sometimes combined with one or more co-stimulatory domains (CD28, 4-1BB,etc.). Please be more precise or paraphrase.
Methods:
Well written, figure 1 helps for better understanding. One could argue to include some subheadings for better overview and understanding of the general concepts of your models, followed by the specifics used in this manuscript.
Results:
Results are well written and presented in an understandable manner.
Discussion:
Well done, the discussion summarizes your results and puts them into context of other in silico and in vitro/in vivo models. The aspect of antigen negative tumor cells as a "barrier" for deeper tumor infiltration of CAR T-cells is very interesting. Most preclinical models focus on the immunosuppressive tumor microenvironment as a hurdle for infiltration and persistence of CAR T-cells. The notion of targeting both, the general tumor cell population as wells as tumor stem cells is interesting but difficult to achieve. One could imagine to use Syn-notch CAR T-cell circuits that first antigen targets on tumor cells and in a later step target tumor stem cells.
Ultimately, your work tries to explain why CAR T-cell therapies currently still fail in solid tumors, e.g. 1)intratumoral antigen heterogeneity 2) antigen escape 3) failing CAR T-cell infiltration into solid tumor tissue maybe due to tumor microenvironment/antigen negative tumor cells. Future model could incorporate more features such as multivalent CAR T-cells, modeling Syn-notch circuits, if possible.
